THE NATURAL HISTORY OF MODEL ORGANISMS

# Neurogenomic insights into the behavioral and vocal development of the zebra finch

**Abstract** The zebra finch (*Taeniopygia guttata*) is a socially monogamous and colonial opportunistic breeder with pronounced sexual differences in singing and plumage coloration. Its natural history has led to it becoming a model species for research into sex differences in vocal communication, as well as behavioral, neural and genomic studies of imitative auditory learning. As scientists tap into the genetic and behavioral diversity of both wild and captive lineages, the zebra finch will continue to inform research into culture, learning, and social bonding, as well as adaptability to a changing climate.

**MARK E HAUBER\*, MATTHEW IM LOUDER AND SIMON C GRIFFITH**

## Introduction

The zebra finch *Taeniopygia guttata* is the most intensively studied species of bird that is maintained in captivity in large numbers despite not being a species bred for its meat or eggs, like the chicken or the quail (reviewed in *Zann, 1996*). It became popular as a pet bird in the 19th century because it bred well in captivity, and was adopted for scientific study in the third quarter of the 20th century, initially for research into sexual behaviors (*Morris, 1954*; *Immelmann, 1972*). Later, the zebra finch was used in studies of the de novo evolution of vocal culture (e.g. *Fehér et al., 2009*; *Diez and MacDougall-Shackleton, 2020*), the neuroethology of imitative vocal learning (*Terpstra et al., 2004*; *Vallentin et al., 2016*; *Yanagihara and Yazaki-Sugiyama, 2019*), the neural mechanisms of sensorimotor learning (*Mandelblat-Cerf et al., 2014*; *Okubo et al., 2015*; *Mackevicius et al., 2020*; *Sakata and Yazaki-Sugiyama, 2020*), and the role of early acoustic experience on the song-based preferences of female mate choice (*Riebel and Smallegange, 2003*; *Chen et al., 2017*; *Woolley, 2012*; see the following video for a mating display in zebra finches: https://www.youtube.com/watch?v=TaC6D1cW1Hs).

Due to the pronounced sexual differences in singing and plumage coloration found in the zebra finch (*Figure 1*), earlier research quickly focused on when and how males learn to copy and produce a tutor(-like) song (e.g. *Eales, 1987*; *Brainard and Doupe, 2002*; *Figure 2A*), and then eventually on how females learn from their (foster) fathers to prefer particular male vocal displays (*Braaten and Reynolds, 1999*; *Riebel, 2000*). This allowed for the characterization and testing of the functions of male song and its female perception in the context of acoustic sexual dimorphism at the behavioral, endocrine, and neurophysiological levels (reviewed in *Riebel, 2009*; *Hauber et al., 2010*).

The zebra finch was the second avian species to have its genome sequenced (*Warren et al., 2010*), after the domestic fowl (*Gallus gallus*; *International Chicken Genome Sequencing Consortium, 2004*). Soon after the appearance of transgenic lines of domestic fowl and the Japanese quail *Cortunix japonica* (reviewed by *Sato and Lansford, 2013*), the first generations of transgenic zebra finches become available (e.g. *Agate et al., 2009*; *Abe et al., 2015*; *Liu et al., 2015*). The proven feasibility of genome editing in both developing zebra finches (e.g. *Ahmadiantehrani and London,*

**\*For correspondence:**
mhauber@illinois.edu

**Competing interests:** The authors declare that no competing interests exist.

**Figure 1.** Adult zebra finches in the wild. Four female and nine male adult zebra finches in the wild in Australia. As the species experiences increasingly extreme climatic fluctuations, future field studies of the zebra finch should also advance our understanding how opportunistically breeding species are able to adapt to accelerating climate change (photo credit: Simon C Griffith).

*2017*) and adult poultry (reviewed in *Woodcock et al., 2017*), means that this bird may also be used as both a basic and an applied (i.e., biomedical) model for development and for human health and disease (e.g. *Han and Park, 2018*; *London, 2020*).

Studies of zebra finch natural history in Australia have been essential to establish and confirm the rationale for studying this species as a model for acoustic communication (*Zann, 1990*; *Elie et al., 2010*), social behavior (*McCowan et al., 2015*; *Brandl et al., 2019a*; *Brandl et al., 2019b*), reproductive physiology (*Perfito et al., 2007*), life-long pair bonding (*Mariette and Griffith, 2012*), and adaptations to heat (*Cade et al., 1965*; *Cooper et al., 2020a*; *Cooper et al., 2020b*). Specifically, by understanding the natural history of the zebra finch, research in captivity can capitalize on the manipulation of the behavioral, neuroendocrine, and epigenetic bases of the bird's phenotype, including conspecific brood parasitism, parent-offspring conflict, and sibling rivalry.

Finally, with Australia experiencing increasingly extreme climatic events and fluctuations, field studies of the zebra finch are also paving the way to understanding how this opportunistically breeding species is adapting to accelerating climate change. For example, recent wild studies have revealed the zebra finch's extensive behavioral and physiological plasticity to withstand extreme temperatures of over 40℃ (e.g. *Cooper et al., 2020a*; *Cooper et al., 2020b*; *Funghi et al., 2019*). In turn, studies of captive zebra finches in controlled temperature conditions have already tested the effects of cool vs. hot climates on parental investment (*Nord et al., 2010*), parent-offspring embryonic communication (*Mariette and Buchanan, 2016*), offspring development (*Wada et al., 2015*), tutor choice for song learning (*Katsis et al., 2018*), adult phenotype (e.g. body size: *Andrew et al., 2017*), the level of DNA methylation (*Sheldon et al., 2020*), and the effect of heat waves on sperm (*Hurley et al., 2018*).

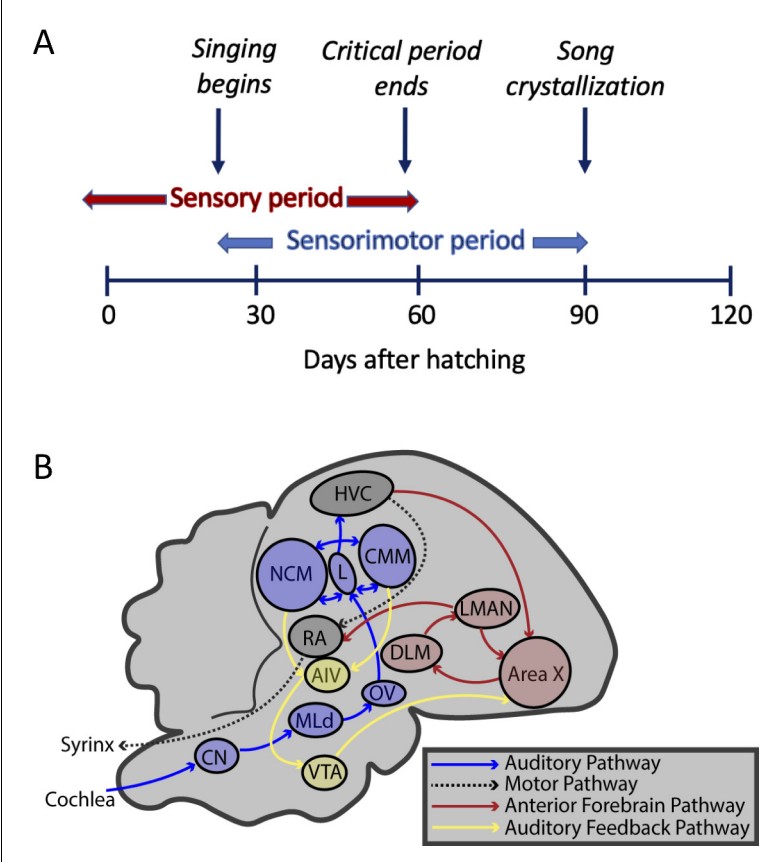

**Figure 2.** Timeline and brain pathways of auditory and vocal learning in the zebra finch. (A) Timeline of sensory (auditory learning) and sensory-motor (vocal self-assessment and song-production) critical periods in zebra finch song development. (B) Brain nuclei of male zebra finches for auditory learning (CN: cochlear nucleus; MLd: mesencephalicus lateralis pars dorsalis; OV: nucleus ovoidalis; field L: primary auditory forebrain input area; NCM: caudomedial nidopallium; CMM: caudomedial mesopallium; VTA: ventral tegmental area; and AIV: ventral portion of the intermediate arcopallium), vocal learning (HVC, Area X: basal ganglia; LMAN: lateral magnocellular nucleus of the anterior nidopallium; DLM: nucleus dorsolateralis anterior thalami, pars medialis), and vocal production (HVC, and RA: robust nucleus of the arcopallium).

By tapping into the existing genetic and behavioral diversity of wild and captive lineages in zebra finches (e.g. *Forstmeier et al., 2007*; *Knief et al., 2015*) to perform comparative avian genomic analyses (*Jarvis et al., 2014*; *Feng et al., 2020*), interspecific hybridization studies (*Woolley and Sakata, 2019*; *Wang et al., 2019*), and direct genetic manipulations (*Liu et al., 2015*; *London, 2020*), the zebra finch shall continue to serve as a focal subject of integrative research into human language-like vocal culture (*Hyland Bruno et al., 2021*), auditory learning (*Theunissen et al., 2004*), acoustically-mediated social bonding (*Tokarev et al., 2017*), and genetic (*Balakrishnan et al., 2010*) and behavioral (e.g. song) variability (*Lansverk et al., 2019*; see *Box 1*).

## An evolutionary history of the zebra finch

The zebra finch is endemic to Australasia, and evolved there as part of the Australian grass finch radiation within the Estrildidae (*Olsson and Alström, 2020*). The species shares a common ancestor with *Poephila* finches (long-tailed *P. acuticauda*; black-throated *P. cincta*; and masked finch *P. personata*), diverging around 2.9 million years ago (*Singhal et al., 2015*). Formerly, the zebra finch was placed in a genus with the double-barred finch (*Taeniopygia bichenovii*), but in fact these two lineages diverged around 3.5 million years ago (*Singhal et al., 2015*).

Two subspecies of the zebra finch are recognized, with the continental Australian taxon (*T. guttata castanotis*) having no clear genetic structure and apparently mating randomly within its breeding population (*Balakrishnan and Edwards, 2009*). The other subspecies is the Timor zebra finch (*T. g. guttata*), found to the north of Australia. The genetic divergence between the two lineages suggests that the latter taxon colonized the Lesser Sunda Islands around 1 million years ago and has a reduced diversity and genetic distance driven by founding effects and selection, relative to the continental subspecies (*Balakrishnan and Edwards, 2009*). The insular subspecies has also been occasionally studied in captivity, and it differs from the continental Australian subspecies in morphological and behavioral traits, including song rate and mate choice (*Clayton, 1990*; *Clayton et al., 1991*).

The two subspecies of the zebra finch are physically isolated from one another in the wild, but they can readily hybridize and be back-crossed in captivity to examine a range of questions in classical genetics and functional developmental biology. To date, this approach has seen limited application, with just one study looking at the divergence in gene regulation between the two subspecies (*Davidson and Balakrishnan, 2016*). Whilst this direction could provide an extremely valuable new research opportunity, a major logistical challenge to overcome will be the capture and export of birds from Indonesia, or the continued maintenance of distinct (non-hybrid) domesticated populations of *T. g. guttata* in captivity.

## Box 1. Outstanding questions in zebra finch research.

- Female zebra finches do not sing but have a diverse repertoire of cooperative calls and other social behavioral displays. What is the neurogenomic and ontogenetic basis of this lack of singing in females?
- Can gene editing become standard practice in both ontogenetic and adult-onset manipulations of the genomic architecture and gene activational basis of focal zebra finch traits, including imitative song learning and auditory feedback in the maintenance of crystallized song production?
- What is the genomic and transcriptomic mechanism of hair-cell regeneration in the songbird inner ear and can it be transferred to human hearing loss treatments?
- What is the genomic and physiological basis of aseasonal reproduction in nomadic zebra finches?

## A model species for the analysis of sex differences in vocal learning and production?

Zebra finches have a relatively short generation time for altricial birds (those that are underdeveloped at the time of hatching): they become sexually mature at between 90 and 100 days of age in captivity, at which point they are ready to form pair bonds, build nests, and breed (*Zann, 1996*). They are highly social and can be kept at great densities in shared housing with a relative absence of highly antagonistic behaviors. This is likely to be related to the level of sociality and the highly fluid flock-wide social relationships seen in the wild (*McCowan et al., 2015*; *Brandl et al., 2019a*), as individuals congregate around food and water, and nest in close proximity in loose colonies for apparent social benefits (*Brandl et al., 2019b*).

Provided with sufficient water, nesting sites, and nest materials, and one (or more) mate(s) of the opposite sex, zebra finches can successfully reproduce on a predominantly seed-based diet, simplifying husbandry, even during the nestling stage. Indeed, under a broad range of environmental and social conditions in captivity, when given the infrastructure (e.g. nesting platform or cavity and materials) to breed, most pairs will breed successfully within a short time frame (*Griffith et al., 2017*), and the life history can be followed across many generations in a relatively short period of time (e.g. *Briga et al., 2019*).

With a clutch size of between 2 and 9 eggs (mode: 5), and with brood reduction rates that can be less than 30%, each reproductive bout is typically rapid and productive. In the wild, zebra finches pair for life, and partners are found in close proximity during both the breeding and non-breeding periods (*Mariette and Griffith, 2012*; *McCowan et al., 2015*). In captivity, this strong pair bond is preceded by rapid pairing, with singletons forming pair bonds within days or weeks when introduced into a new cage or aviary (*Rutstein et al., 2007*; *Campbell et al., 2009*). The strength of the pair bond, the high levels of affiliative behaviors, and the relative absence of antagonism between partners also allow zebra finches to be kept in easily monitored single-pair cages, rather than in communal aviaries (*Zann, 1996*).

However, it was not just ease of breeding in captivity that turned the zebra finch into a popular model for studying the development of sexual dichromatism and vocal dimorphism. Rather, an initial interest in the distinct plumage and the vocal differences between adult female (drabber, non-singing) and male (more colorful, singing) zebra finches resulted in several, now classic, developmental studies. Some of these studies concentrated on the role of early life experience, through chromatic and vocal sexual imprinting, on females choosing attractive males as mates, while others focused on song production and song preference learning by male and female zebra finches (e.g. *Clayton, 1987*; *Eales, 1987*). For example, cross-

fostering zebra finch chicks with the 'universal estrildid foster species', the Bengalese finch (*Lonchura striata vars. domestica*; *Sonnemann and Sjölander, 1977*), revealed that both visual and acoustic cues of social parents are learned during early development and used by young zebra finches of both sexes in mate preference following maturity (*ten Cate, 1987*; *Campbell and Hauber, 2009*; *Verzijden et al., 2012*). This occurs through a two-stage process of sexual imprinting (*ten Cate, 1985*; *ten Cate and Voss, 1999*).

These ontogenetic, physiological, and behavioral studies since the last quarter of the 20th century (e.g. *Price, 1979*) have become increasingly coupled with the rapid advances of neuro-anatomical and neurophysiological imaging, genome sequencing, and transcriptomic and epigenetic analyses of the neural circuitries of song production in the forebrains of songbirds (reviewed in *Mooney, 2009*; *Mooney, 2014*) and song perception (reviewed in *Louder et al., 2019*). For instance, neurophysiological (*Hauber et al., 2013*), neuroanatomical (*Lauay et al., 2005*), immediate-early gene (*Tomaszycki et al., 2006*), and transcriptomic analyses (*Louder et al., 2018*) performed on zebra finch females that were reared either in isolation from any male birdsong or in the presence of a different songbird species have confirmed the critical role of early life experience in generating adaptive cognitive-behavioral (*Price, 1979*), neurogenomic (*Louder et al., 2018*) and neurophysiological (*Moore and Woolley, 2019*) responses to conspecific songs. Similarly, the known upregulation of stress responses of formerly pair-bonded, but then separated captive zebra finches (*Remage-Healey et al., 2003*), is also reported to impact the epigenomic status of similarly treated birds (*George et al., 2020*).

Despite the earlier prominence of the domestic canary (*Serinus canaria*) in the neurobiological study of song learning, two other research themes have also benefited significantly from follow-up studies of captive zebra finches. First, adult-onset neurogenesis, accompanying seasonal changes in song behavior, or damage to the underlying neural circuitry, was initially extensively studied in the canary (e.g. *Notte-bohm, 1981*), but with ongoing critical contributions also coming from experiments on zebra finches (e.g. *Walton et al., 2012*; reviewed in *Pytte, 2016*). For example, when adult male zebra finches' RA- (robust nucleus of the arco-pallium) and Area X-projecting HVC neurons (*Figure 2B*) were experimentally ablated, only the RA-projecting neurons were regenerated (*Scharff et al., 2000*). In turn, a new social environment (e.g. through the exposure to novel aviary mates: *Barnea et al., 2006*, and/or ongoing auditory experiences: *Pytte et al., 2010*) may also contribute to the diminished apoptosis of newly generated caudomedial nidopallium (NCM) neurons (*Figure 2B*) in the forebrains of adults.

Second, hair cell regeneration following a loud noise or antibiotic treatment in both Bengalese (*Woolley and Rubel, 2002*) and zebra finches (*Dooling and Dent, 2001*) occurs rapidly, as it does in other, non-oscine birds (*Stone and Rubel, 2000*) and in some other vertebrate lineages (e.g. fish: *Monroe et al., 2015*). Research into such auditory system regeneration abilities in birds and other animals had strongly promised, but has thus far evaded, broadly applicable biomedical solutions for curing cell-death based hearing losses in humans (*Brigande and Heller, 2009*; *Menendez et al., 2020*).

## Differences in captive vs. wild zebra finches and comparisons with northern hemisphere songbirds

Most of the populations of zebra finches in research laboratories around the world have been founded with birds held by aviculturists for over a hundred generations (*Zann, 1996*; *Griffith et al., 2017*). These populations have therefore been subject to both direct and indirect forms of natural and artificial selection, as well as founding effects, genetic drift, and inbreeding (*Forstmeier et al., 2007*; *Knief et al., 2015*). It has long been known that birds of the domesticated stocks are up to 30% larger in body size than their wild counterparts (*Zann, 1996*), but reassuringly they appear to be similar with respect to several life history trade-offs, including, for example, slow juvenile feather development and low adult song rates when nestlings are raised in large brood sizes (e.g. *Tschirren et al., 2009*). Captive birds are also similar to their wild counterparts in respect to the genomic architecture underlying complex traits (*Kim et al., 2017*; *Knief et al., 2016*; *Knief et al., 2017a*), although some caution still needs to be applied, for instance, to known differences in linkage disequilibrium patterns within the genomes of captive and wild populations (*Knief et al., 2017b*).

The pattern of zebra finches being quite different from many of the species of small passerines that are well studied by researchers in the northern hemisphere may be of greater significance than the differences between captive and wild populations of zebra finches. The zebra finch is an estrildid (*Sorenson et al., 2004*; *Olsson and Alström, 2020*), a family that is endemic to the tropics, and found across Africa, Southern Asia, and Australasia – with the whole lineage having evolved far from the ecological and evolutionary pressures of the temperate northern hemisphere. One of the almost ubiquitous characteristics of the estrildid family is the interseasonal strength of the socially monogamous pair-bond and biparental care for the young (*Payne, 2010*).

Prior breeding experience enhances the success of subsequent breeding bouts by female zebra finches through increased output and shorter times between clutches, even when breeding with a new male in this otherwise lifetime pair-bonded species (*Adkins-Regan and Tomaszycki, 2007*; *Smiley and Adkins-Regan, 2016*; *Hurley et al., 2020*). Relatively high within-pair sexual fidelity and cooperation in nest building, incubation, and provisioning also allow for the directed breeding of known pairs both in large aviaries and in small single-pair cages. Nevertheless, in socially housed groups, both conspecific brood parasitism – inducible by simulated nest predation in captivity (*Shaw and Hauber, 2009*) and accounting for 5 to 11% of offspring (*Griffith et al., 2010*) – as well as extra-pair paternity – accounting for around 30% of offspring in aviaries (*Forstmeier et al., 2011*) – can partially confound social parentage, although extrapair paternity is almost entirely absent in the wild (accounting for ~1% of offspring; *Griffith et al., 2010*).

A major effort of laboratory-based work on the zebra finch has focused on females' mate choices (especially with respect to beak color and learned song; *Griffith and Buchanan, 2010a*). However, despite considerable variance in the reproductive success of individuals even in captive populations (*Griffith et al., 2017*; *Wang et al., 2017*), one of the most comprehensive studies examining the consequence of mate choice on fitness found no evidence that either males or females are targeting this variation in individual quality when they choose a partner (*Wang et al., 2017*). This finding supports the idea that the strength of a partnership is of greater value than the intrinsic quality of the individuals involved.

In this respect, zebra finches may differ from similarly-sized well studied small passerines of the northern hemisphere temperate zone. Since adult zebra finches are likely to live between 3 and 5 years in the wild (*Zann, 1996*) and can breed continuously throughout the year if conditions are favorable (*Griffith et al., 2017*), they can potentially accrue considerable experience as part of the sexual-parental partnership. The reproductive benefits of better physiological and behavioral coordination between partners (e.g. *Adkins-Regan and Tomaszycki, 2007*; *Smiley and Adkins-Regan, 2016*; *Hurley et al., 2020*) may outweigh the benefits of frequent and repeated partner switching and genetic infidelity (*Griffith, 2019*). In turn, the value of the partnership may promote selection for diverse affiliative and cooperative traits, not always seen in the widely studied passerines of the more seasonally constrained northern hemisphere, where most individuals breed just once or twice in a lifetime (*Griffith, 2019*). Rather, these traits are reminiscent of the long-term cooperative breeding partnerships formed (and the fitness costs paid following divorce or mate loss) by long-lived biparental seabirds (e.g. *Ismar et al., 2010*).

Indeed, the strength of the pair bond in the wild zebra finch is seen in the expression of acoustic communication throughout the year, and high levels of coordinated duetting between the male and female (*Elie et al., 2010*). This close, and regular vocal interaction between the members of a pair also perhaps plays a role in individual vocal recognition in this species (*Levréro et al., 2009*; *Elie and Theunissen, 2018*; *Yu et al., 2020*).

Highly coordinated acoustic interactions between female and male partners are a characteristic of the earliest passerine lineages as they had evolved in Australia (*Odom et al., 2014*). The continuously high level of overall acoustic activity in the zebra finch, which has made it such an attractive model system for neurobiology, sets it apart from many other well studied passerines in the northern hemisphere. This serves to remind us that although most of the laboratory work is conducted in the northern hemisphere, the zebra finch is, in many respects, different from most of the short-lived highly seasonally breeding passerines native to the temperate zone of the northern hemisphere. Indeed, it is important to understand that the species' adaptations to the highly unpredictable Australian climate and ecology – while making it so easy to maintain and breed in captivity – also

set it apart from most other northern hemisphere lineages that could not be used in laboratories to anywhere near the same extent.

## Genes and brains for vocal learning

The process through which developing young memorize the acoustic communication signals of adults in humans and songbirds has been a critical research rationale and funding source supporting zebra finch studies. The learning of adult male songs by juveniles is particularly strong during early sensory periods, when embryos (*Antonson et al., 2021*), nestlings (*Rivera et al., 2019*), and juveniles (*Brainard and Doupe, 2000*) likely form a sensory representation of the 'tutor song' (*Figure 3*). Just as juvenile females develop long-term song-type preferences used for mate choice based on early experiences with their own fathers (*Riebel, 2000*; *Chen et al., 2017*), young males also learn and then actively practice to produce songs that match their paternal (tutor) songs (*Tchernichovski et al., 2001*; *Figure 3*). Tutors even alter their song structure when singing near young tutees, which influences the song learning process for young zebra finches, analogous to humans changing their speech when speaking to infants (*Chen et al., 2016*; *Carouso-Peck and Goldstein, 2019*).

However, even in the case of strong social environmental impact upon song learning during the sensitive period, the genetic make-up of individuals may contribute to the resulting song preferences and vocal production patterns through gene-by-environment interactions (*Mets and Brainard, 2019*). Accordingly, in zebra finches, males preferentially learn to sing from song tutors of the same species over those of another species when given equal access (*Clayton, 1988*), and both song-naïve and cross-fostered females show greater neuronal spike rates in response to unfamiliar conspecific over an unfamiliar third species' songs (*Hauber et al., 2013*). Similarly, the species-specific typical pattern of socially learned song structure can culturally evolve across of just a handful of generations in initially naïve zebra finch populations (*Fehér et al., 2009*; *Diez and MacDougall-Shackleton, 2020*).

In adulthood, male and female zebra finches can quickly memorize individual vocal characteristics and recognize the identity of others for at least a month without reinforcement (*Yu et al., 2020*), likely relying on the perception of extremely small differences in calls and songs (*Prior et al., 2018*). However, experiences with other songs in adulthood do not affect the crystallized songs of males. Given the parallels with language acquisition and speech development in humans, zebra finches have thus long served as an important model for studying the neural mechanisms that control how vocal signals are memorized and copied (*Doupe and Kuhl, 1999*).

Initial research in the neurobiology of songbirds, primarily with canaries, has revealed the components and plasticity of the neural loops and circuits responsive to learning and producing songs (*Figure 2B*). Over time, studies of the zebra finch (a species that crystallizes its specific song once and does not deviate from it unless experiencing trauma or training) have become increasingly more instructive in the pursuit of identifying where in the forebrain the auditory memories are stored and how this representation directs both vocal learning in males and mate choice preferences in females (reviewed in *Hauber et al., 2010*). Accordingly, following the presentation of tape-recorded songs of conspecifics, the expression level of an immediate-early gene, egr-1 (also known as ZENK), which is associated with neural activation, increases within the zebra finch auditory forebrain, as found in other songbird species (*Mello et al., 1992*; *Louder et al., 2016*).

Furthermore, neural responses within the NCM, a subregion of the auditory forebrain, are selective for tutor songs (*Yanagihara and Yazaki-Sugiyama, 2016*) and song-induced expression of neural transcription factors (again, ZENK) also positively correlate with the increased similarity of the bird's copied song to that of the tutor (*Bolhuis et al., 2000*), which together suggest that this region may hold the tutor song's memory. Accordingly, NCM lesions in adult male zebra finches reduce their ability to recognize songs, but not to produce them (*Gobes and Bolhuis, 2007*). In female zebra finches, on the other hand, behavioral preferences for conspecific versus heterospecific songs can be eliminated by damaging the nearby CMM nucleus (caudomedial mesopallium) (*MacDougall-Shackleton et al., 1998*).

Overall, the zebra finch remains the best model system to characterize the neural circuitry involved in vocal learning and production, with an often-stated research aim to better understand the capacity of imitative speech learning in humans (e.g. *Lipkind et al., 2013*). Juvenile male zebra finches mimic the tutor song while

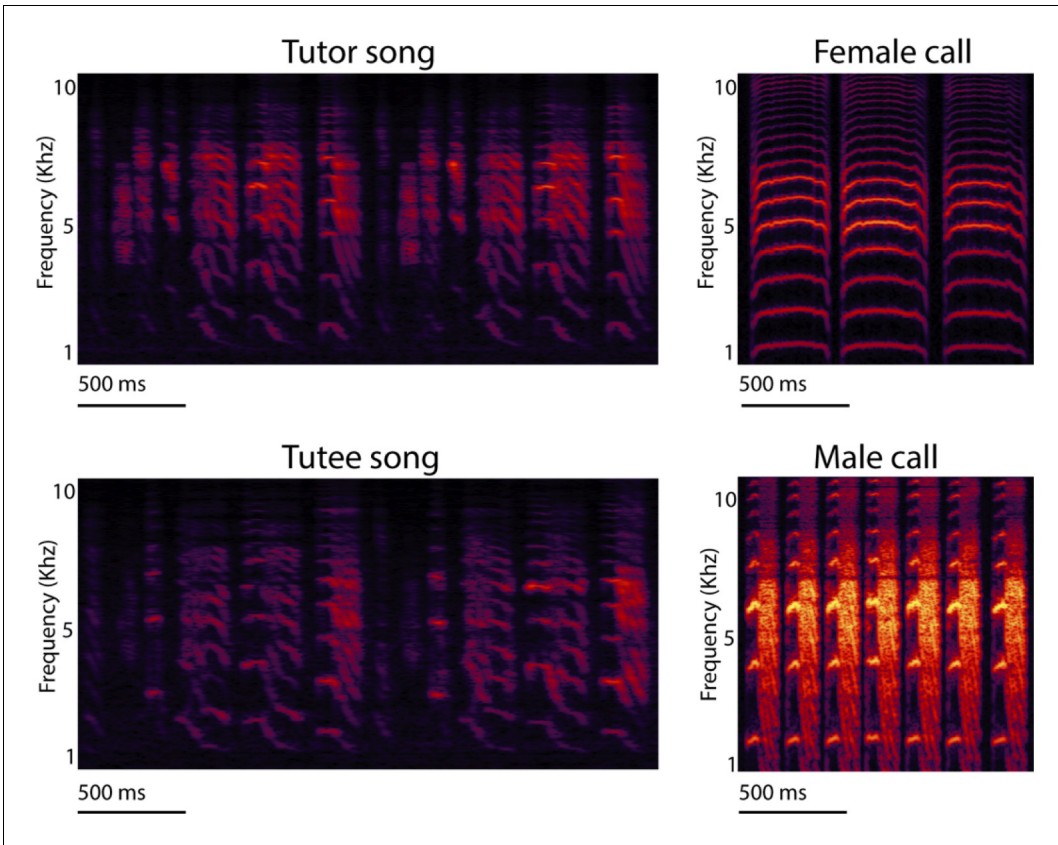

**Figure 3.** Spectrograms of zebra finch songs and calls. Spectrogram of tutor and tutee adult male zebra finch songs, and undirected contact calls of adult females and males. Spectrograms represent time (x-axes) and pitch (y-axes) with greater amplitude as increasing brightness. Note the similarity of the tutor (typically social father) and tutee (son) song pair of male zebra finches and the distinct sexual differences of the calls.

females only produce non-learned 'calls' (*Figure 3*). In turn, several regions in the zebra finch brain associated with song production are dramatically larger in male zebra finches, a result of neurons in some of these regions atrophying in females while increasing in size and connections in males (*Figure 2B*; *Konishi and Akutagawa, 1985*). Several of these regions selectively respond to the 'bird's-own-song' in anesthetized males (*Doupe and Konishi, 1991*), which initially suggested a specialized function for this circuit in producing songs; however, the role of such own-song specific auditory responses is no longer clear, as they are gated by behavioral states (*Hessler and Doupe, 1999*) and much less pronounced in awake birds (*Schmidt and Konishi, 1998*).

The premotor circuit for song production receives input from auditory nuclei via the HVC, which then projects to the RA, and subsequently connects to the brainstem motor nuclei and syrinx (*Figure 2B*). This 'motor pathway' is crucial during the learning process (*Aronov et al.,*

*2008*) to generate stereotyped adult songs (*Simpson and Vicario, 1990*). In turn, while singing, neurons in the HVC that connect to the robust nucleus of the arcopallium (RA) perform time-locked bursts of firing, coincident with precise sequences during the song (*Hahnloser et al., 2002*). HVC neurons also ontogenetically shift their spike rates to become increasingly sparser while producing the male's song (*Okubo et al., 2015*), whereas the spike trains of RA neurons lock into the timing of song's note identity (*Ac and Margoliash, 2008*). By altering the local temperature of specific brain nuclei, *Long and Fee, 2008* demonstrated that the temporal match between HVC, but not RA, and the song's timing pattern is a causal link, as cooling the HVC, but not the RA, slows down the song without affecting its frequency content. This demonstrates how and which elements of this forebrain circuit are critical to controlling the temporal structure of male songs and, in the Bengalese finch, their syntax, too (*Zhang et al., 2017*). By contrast, the anterior

forebrain pathway (AFP), homologous to the mammalian basal ganglia–thalamocortical pathway, is required for vocal learning in juvenile male zebra finches, but not the production of stereotyped adult song (*Bottjer et al., 1984*). In this pathway, Area X and the lateral magnocellular nucleus of the anterior nidopallium (LMAN) are involved in producing song variability in juvenile birds during vocal learning (*Woolley and Kao, 2015*; *Figure 2B*).

Specifically, both theoretical modelling (including in humans) and experimental studies of this pathway (in zebra finches) have pointed to the critical role of vocal motor variability as the substrate upon which trial-and-error learning through reinforcement mechanisms may operate to shape vocal production ontogeny (*Dhawale et al., 2017*). In turn, the AFP is also involved in auditory-feedback based acoustic correction signaling for the motor pathway, in that inactivation of LMAN in young male zebra finches regresses experimentally induced, recently learned changes in the subjects' song pitch (*Andalman and Fee, 2009*). Finally, gene expression patterns, including genes associated with speech in humans such as the transcription factor FOXP2, are highly expressed in the anterior forebrain pathway during sensitive periods for song learning, indicating potential genetic parallels of vocal plasticity in birds and humans (*Haesler, 2004*; *Pfenning et al., 2014*).

How the memorized tutor song instructs vocal pathways remains unclear. However, research in the zebra finch points to the involvement of nuclei within and outside of the anterior forebrain pathway. Auditory feedback, in which self-uttered and self-heard vocalizations are compared to a memorized song pattern, is necessary for the development of song in juveniles and the maintenance of song in adult zebra finches (*Price, 1979*; *Nordeen and Nordeen, 1992*; *Leonardo and Konishi, 1999*). Dopaminergic neurons of the ventral tegmental area (VTA) that project to the anterior forebrain pathway through Area X encode perceived errors in song performance from auditory feedback (*Gadagkar et al., 2016*; *Figure 2B*). The VTA receives error signals from auditory feedback through the AIV, which receives connections from the auditory forebrain (*Kearney et al., 2019*). Furthermore, neurons within the auditory forebrain also demonstrate sensitivity to errors in auditory feedback (*Keller and Hahnloser, 2009*). Such developments, for example regarding error sensitivity, also illustrate how ongoing research and continued breakthroughs in zebra finch neuroscience hold promise to further identify and understand the neural basis of vocal learning and production in general.

Following the widespread use of immediate early gene studies (see above), some of the research efforts aiming to characterize the genes that regulate zebra finch vocal and auditory behaviors, in particular genes related to vocal production in the brain, were based on utilizing DNA microarrays (*Wada et al., 2006*). Then, in 2010 an international consortium sequenced, assembled, and annotated the first zebra finch genome (*Warren et al., 2010*), only the second avian genome presented. This effort revealed the sequences of over 17,000 predicted protein-coding genes, as well as many regulatory regions and non-coding RNAs. More importantly, the annotated genome enhanced the next decade's analyses into identifying the genes and regulatory networks that are involved in social behavior, including genome-wide investigations into vocal learning, such as auditory-experience induced RNA expression (*Louder et al., 2018*), microRNA expression (*Gunaratne et al., 2011*), and epigenetically regulated genes associated with developmental song learning (*Kelly et al., 2018*). Furthermore, the initial genome helped researchers to identify and map the expression patterns of ~650 candidate genes within the brain of zebra finches, resulting in an online atlas database that provides an opportunity to link behavior, neuroanatomy, and molecular function (*Lovell et al., 2020*).

A recent high quality, second generation genome of the zebra finch, presented as part of the Vertebrate Genomes Project, improves the accuracy of the reference genome assembly and annotation (*Rhie et al., 2021*). Leveraging recent technological advances, such as long-read sequencing (up to 100 Kbp) and approaches to detect how DNA interacts across genomic loci (up to 100 Mbp), the latest updated zebra finch genome thus resolves numerous regions with repetitive elements and enhanced gene annotation from the first assembly.

In parallel with genomic advances, a suite of new neurobiological techniques available for zebra finches will only continue to increase the ability to understand the development of vocal learning and behavior. Questions regarding the activity of specific neurons can now be tackled using multi-electrode arrays (e.g. *Lim et al., 2016*; *Tanaka et al., 2018*) or wireless neurotelemetry (*Ma et al., 2020*) able to simultaneously record the activity of numerous neurons in

awake and freely-behaving birds. Imaging the neural connections between distant brain regions is now also possible with tissue clearing and light-sheet microscopy (*Rocha et al., 2019*).

The experimental regulation of the expression of candidate genes in targeted areas of the zebra finch brain has also recently become available. Existing or new gene constructs can be inserted into neonatal (hatchling) zebra finches via electroporation-based gene construct delivery to study the genetics of vocal learning as songs are memorized, practiced, and first expressed by young males (*Ahmadiantehrani and London, 2017*). Similarly, genetically modified constructs of nonpathogenic viruses injected in the brain, such as adeno-associated virus (AAV), are able to drive the expression of certain genes.

Viral constructs were developed to control the expression of FOXP2 (e.g. *Heston and White, 2015*; *Norton et al., 2019*), which is expressed in the song control regions within the male zebra finch forebrain and associated with inherited speech and language disorder in humans (*Fisher and Scharff, 2009*). Viral constructs have also been useful in imaging, such as expressing a genetically encoded calcium indicator (GCaMP6s) for calcium imaging of neuron populations with 2-photon microscopy (*Picardo et al., 2016*) or the expression of green fluorescent protein (GFP). Recent applications of viral constructs have also enabled researchers to control neurons with light (optogenetics), such as 'implanting' artificial song memories into the zebra finch brain (*Zhao et al., 2019*), or controlling the firing of specific neurons, such as the VTA neurons that project to Area X (*Xiao et al., 2018*; *Kearney et al., 2019*). Harnessing these new techniques enables us to tackle how genetic pathways are linked to vocal learning and motor control circuits.

However, the utility of the zebra finch as a neurogenetic model laboratory species has been somewhat inhibited by the low success rate in the development of transgenic lines that would enable direct experimental modification of the gene expression patterns in the relevant vocal-production and vocal-perception circuits. This may be due to the unique immune function of oscine birds inhibiting full viral delivery of gene constructs (*London, 2020*). Nevertheless, the last decade has already seen the successful innovation of lentiviral delivery (e.g. *Norton et al., 2019*) of, for example, human Huntington's Disease genes into zebra finch lineages, to causally demonstrate reduced vocal imitation and output

consistency as a result of the treatment (*Liu et al., 2015*). However, to date neither a TALEN nor a CRISPR/Cas9 vector-based gene editing approach has taken off in avian (chicken or songbird) lineages (*Woodcock et al., 2017*; but see *Cooper et al., 2018*). With additional research, the zebra finch could be further explored as to which gene delivery and genomic editing methods will be widely and effectively applicable to this species.

## The importance of studying female zebra finches

Female zebra finches only slowly and partially assumed a role in some of the earlier behavioral and developmental studies on sexual imprinting (e.g. *Collins et al., 1994*), but now maintain a co-lead position. This is because mate choice is mutual in this species and females participate in the ever-important initial pair-bonding decisions, as well as in all aspects of collaborative biparental care (*Riebel, 2009*). As such, females make a critical contribution to the phenotype of their offspring through their investments into eggs, and the care of dependent offspring (*Griffith and Buchanan, 2010b*). Still, in studying the neurobiological basis of species and mate recognition, and the relevant funding and publications, female-focused research took a secondary role during the earlier decades when much of the work focused on the developing and adult sensory-motor circuitries of the male zebra finch forebrain.

In the last two decades, however, there has been a definite upsurge of studies focusing on female zebra finches, both from the perspective of the neurosensory-ontogenetic processes of conspecific (*Theunissen et al., 2004*; *Woolley et al., 2010*), mate (*Lauay et al., 2004*; *Tokarev et al., 2017*), and individual recognition (*Vignal et al., 2004*; *D'Amelio et al., 2017*; *Yu et al., 2020*) by and of females. It is becoming clear that female visual and acoustic displays serve an important role in the development and fine-tuning of male vocalizations during sensitive periods (*Benichov et al., 2016*; *Carouso-Peck and Goldstein, 2019*) and that male vocal and/or visual displays serve in the activation of auditory forebrain regions in adult females (*Avey et al., 2005*; *Day et al., 2019*).

For example, the reduced volume of the song control system that exists in the female zebra finch brain is likely not at all vestigial (*Shaughnessy et al., 2019*) and may be even more functional than previously thought,

enabling plasticity in the vocal timing of calls in social interactions (*Benichov et al., 2016*). In turn, female (and male) parental vocal communication with embryos in ovo in the nest have also been discovered to shape not only the functional neurogenomic responses of the embryos themselves (*Rivera et al., 2019*) but also the acoustic tutor choice of young male zebra finches (*Katsis et al., 2018*), as well as adult behavioral phenotypes and reproductive success (*Mariette and Buchanan, 2016*).

Finally, the behavioral, the neurophysiological and gene-activational bases of perceptual learning of conspecific song features appear to be both species-specific in song-naïve (mother-only parent raised) female zebra finches and dependent on early social experience with con- or cross-fostered heterospecific male songs (*Hauber et al., 2013*; *Louder et al., 2018*). Some of these latter discoveries in females have been made possible through cross-fostering nestling zebra finches with estrildid finch tutors of other species (e.g. *Clayton, 1987*). Critically, the results from females have now also been both replicated and advanced in cross-fostered males. Specifically, the extent of heterospecific song learning in males can be directly measured by the altered songs that they produce following experimental manipulation of early song exposure, and compared with the extent of neurophysiological response selectivity for conspecific (innate) vs. heterospecific (learned) tutor songs and their contributory bioacoustic features in the brain (*Moore and Woolley, 2019*). In turn, cross-fostered males singing the foster species' song famously show an inability to copy the temporal pattern of heterospecific songs, discovered to be due to a lack of ontogenetic flexibility in the neurons that encode heterospecific song-gap (silent period between song bouts) perception again within field L of the auditory forebrain (*Araki et al., 2016*).

## Conclusions

The zebra finch was not originally brought into the laboratory as a model system, nor championed as such by early research pioneers. From the 1950s onwards, the species has been progressively adopted as a useful focus of study in an increasing set of research fields, largely due to its accessibility and the ease with which it can be held and bred in captivity. In contrast, wild passerine birds have long been the focus of ecological and evolutionary research in the northern hemisphere. When studies of free-living study populations were unable to achieve the necessary manipulative rigor, the zebra finch, found commonly in pet shops throughout Europe and North America, became widely adopted as a surrogate captive experimental model. In parallel with its use in early ethological research, the zebra finch became established as an easier model than the canary for studying the neural basis of song, which in turn saw the former species adopted as a model for genomics, neuroscience, and developmental biology.

The zebra finch has provided great insights into diverse fields in biology and has travelled a long path from its natural habitat in arid Australia. It is important to be mindful that the traits that have contributed to its utility and adoption as 'the' avian laboratory model species for basic and biomedical research set it aside from most other avian species. The zebra finch evolved in an austral ecological setting that is profoundly different from those in the many geographic regions where most of this laboratory work takes place.

The zebra finch remains almost uniquely suited as a model system for research and the path ahead is likely to be productive and insightful in established and new areas of research. The late Richard Zann's excellent monograph of the species (1996), whilst already over two decades old, still provides an excellent overview into the natural history of the species, and is never far from our desks, for the insight that it brings. We encourage future adopters of the zebra finch as a research model to use this book to guide their planning and to help interpret their results. The zebra finch is the most widely researched laboratory songbird in the world because of its uniqueness, and not as a result of any advocacy.

## Acknowledgements

We thank our many colleagues working with zebra finches for productive discussions, including the editors and the referees of eLife. M Rivera and O Tchernichovski kindly provided song and call files of adult zebra finches. We dedicate our article to the memories of the late Richard Zann and Mark Konishi. FUNDING Financial support was provided by the Harley Jones Van Cleave Professorship and the National Science Foundation IOS # 1456524 (to MIML and MEH).

**Mark E Hauber** is in the Department of Evolution, Ecology, and Behavior, School of Integrative Biology, University of Illinois at Urbana-Champaign, Urbana-Champaign, United States
mhauber@illinois.edu

https://orcid.org/0000-0003-2014-4928

**Matthew IM Louder** is in the International Research Center for Neurointelligence, University of Tokyo, Tokyo, Japan and the Department of Biology, Texas A&M University, College Station, United States
https://orcid.org/0000-0003-4421-541X

**Simon C Griffith** is in the Department of Biological Sciences, Macquarie University, Sydney, Australia
https://orcid.org/0000-0001-7612-4999

*Author contributions:* Mark E Hauber, Conceptualization, Writing - original draft, Project administration, Writing - review and editing; Matthew IM Louder, Writing - original draft, Writing - review and editing; Simon C Griffith, Conceptualization, Writing - original draft, Writing - review and editing

*Competing interests:* The authors declare that no competing interests exist.

## Funding

| Funder | Grant reference number | Author |
|---|---|---|
| National Science Foundation | IOS1456524 | Mark E Hauber |

The funders had no role in study design, data collection and interpretation, or the decision to submit the work for publication.

### Decision letter and Author response

Decision letter https://doi.org/10.7554/eLife.61849.sa1
Author response https://doi.org/10.7554/eLife.61849.sa2

## Additional files

### Data availability

No new data were generated in this study.

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
