## [Decision Letter]

Thank you for submitting your article "The Natural History of Model Organisms: Neurogenomic insights into behavioral and vocal development of the zebra finch" to *eLife* for consideration as a Feature Article. Your article has been reviewed by three peer reviewers and the evaluation has been overseen by two editors from the *eLife* Features Team (Helena Pérez Valle and Peter Rodgers). The following individual involved in review of your submission has agreed to reveal their identity: Leslie Phillmore (Reviewer #3).

The reviewers and editors have discussed the reviews and we have drafted this decision letter to help you prepare a revised submission.

Summary:

This article reviews the history of the zebra finch as a species of choice to study different aspects of developmental neurobiology and vocal communication. However, the article is lacking references to important works, and should also discuss the roles of the descending motor pathway in relation to the anterior forebrain pathway for song learning and adult song production.

Essential revisions:

1. Please include references to the work from labs that have substantially contributed to the fields of neuroscience and behaviour using zebra finches, including the Fee lab (MIT), the Goldberg lab (Cornell), the Sakata lab (McGill). Also please ensure that the article includes citations to the key works of other labs that are already cited, including the Jarvis lab (Rockefeller), the Long lab (NYU), the Brainard lab (UCSF), the Tchernikovski lab (CUNY) and the Theunissen lab (Berkeley). Please add up to 20 references to address this point. Additionally, please consider the following:

a) Please cite Desmond Morris (1954 and 1957) along with Immelmann when talking about the reproductive behaviour of zebra finches (line 35), and try to find a reference for Immelmann that is not an abstract for a conference.

b) Please cite Katharina Riebel (2000) "Early exposure leads to repeatable preferences for male song in female zebra finches", or any of her later reviews, in addition to Braaten and Reynolds (1999) (line 40).

c) Please check whether Day, 2019 is the correct citation in line 48; and whether work by the White, Mooney, Roberts or Scharff labs would be more appropriate when talking about genome editing.

d) Please check whether there are references missing in line 51 after "illness".

e) Please add references for genetic diversity in line 73 (e.g. "Genetic variation and differentiation in captive and wild zebra finches (Taeniopygia guttata)", from the Forstmeier lab).

f) Please cite reviews on line 140, since the literature is extensive. "Sexual imprinting and evolutionary processes in birds: a reassessment", Ten Cate and Vos (1999), and "The impact of learning on sexual selection and speciation", Verzijden et al. (2012).

g) In line 145, Louder et al. covers auditory but not song production, please find an appropriate reference to cover both.

h) In line 320, please add a reference to Scharff's work on FoxP2

i) In line 334/35, please provide a reference for "the unique immune function of oscine birds inhibiting full viral delivery of gene constructs".

2. Please add between 500 and 1000 words describing the roles of the descending motor pathway vis-à-vis the anterior forebrain pathway for song learning and adult song production; and the use of the zebra finch to study auditory perception of both biologically relevant stimuli (i.e. conspecific vocalizations) and others (e.g. human speech patterns).

3. Figure 2B should include spectrograms of at least a female call (see line 272), perhaps both a male and female call, as well as the male song.

4. The authors' work currently makes up about a third of the references, please check whether these references are all necessary and whether it might be possible to replace some of them with references to other important works in the field.

[Editors' note: further revisions were suggested prior to acceptance, as described below.]

Thank you for resubmitting your work entitled "Natural History of Model Organisms: Neural and genomic insights into behavioral and vocal development of the zebra finch" for further consideration by *eLife*. Your revised article has been reviewed by a new peer reviewer (Mimi Kao), and the evaluation was overseen by Helena Pérez Valle as the Assistant Features Editor and Peter Rodgers as the Features Editor.

The manuscript has been improved but there are some remaining issues that need to be addressed, as outlined below.

Summary:

This article reviews the history of the zebra finch as a species of choice to study different aspects of developmental neurobiology and vocal communication, and particularly as a system for investigating sex differences in vocal communication. The revised manuscript is much improved, but it would still benefit of additional revisions to references, and including a discussion of studies of auditory perception using cross-fostered birds to examine innate preferences versus experience-dependent changes in auditory responses.

Essential revisions:

1. Please make the following modifications to references:

a) In line 37, add references to the work of the Fee and Yazaki-Sugiyama labs. These would be useful for neural mechanisms underlying sensorimotor learning (e.g. Okubo et al., 2015; Mackevicius, Happ, and Fee, 2020; Yanagihara and Yazaki-Sugiyama, 2019).

b) In lines 38-39, the reference to Chen et al., 2017 does not seem to be appropriate, since that study focuses on vocal learning, not female preference. Please remove this reference and replace it with another that examines influences on female song preferences (e.g. Riebel, 2000).

c) In lines 63 and/or 73, please add a reference to Cade et al., 1965, Water economy and metabolism of two estrildine finches, Physiological Zoology.

d) In lines 88-89, please add references to papers by Mets and Brainard (2018, 2019) that investigate the interaction of genes and social and/or auditory experience on vocal

learning and performance. Please add a short discussion about gene-environment interactions that shape song learning based on the findings of these papers in the section "Brains and genes".

e) In line 159 please add a reference to Price, PH (1979) Developmental determinants of structure in zebra finch song. Journal of Comparative and Physiological Psychology.

f) Please check the references in lines 258-260. While several papers have shown that both calls and songs have strong individual signatures that can be used for individual recognition, the references cited did not specifically examine "high levels of coordinated duetting between the male and female".

g) In lines 323-327, please add a reference to Aronov et al. (already cited in the article) in regard to the critical role of HVC for generating stereotyped, learned vocalization in lines 323-327.

h) In lines 330-331, please add references for the idea that the AFP generates song variability in juvenile birds (Olveczky et al., 2005, 2011; or a review article, e.g. Mooney, 2009; Woolley and Kao, 2015; or Dhawale, Smith and Olveczky 2017), and discuss these findings.

i) Please add references to studies showing that LMAN also sends an instructive signal that can bias song and drive systematic changes in mean pitch, mean duration and syllable transition probabilities (Turner and Brainard, 2007; Andalman and Fee, 2009; Ali et al., 2013, Charlesworth et al., 2012), and discuss these findings.

j) In lines 341-344, please add a reference for deafening experiments in juvenile birds, such as Price, 1979.

k) In lines 349-351, it is unclear why the Ma et al., 2020 reference is included, since the paragraph discusses mechanisms by which evaluation of song performance using auditory feedback can drive changes in song. Ma et al., 2020 report activity prior to calls by conspecifics. While predictive activity may be important for learning or maintaining song, please make clear what the activity is predicting (timing of upcoming calls? Expected auditory feedback given the motor commands?) and how it relates to learning the tutor song model.

l) In line 401, please add Xiao and Roberts, Neuron, 2018 as a reference, in addition to Kearney et al., 2019.

m) In lines 441-444, please add a reference to Shaughnessey et al., JCN, 2018 when mentioning the song system in females as that paper argues against the idea that females possess only vestiges of the song control network and suggests that the forebrain network may have a different function in females.

2. Given the manuscript's emphasis on vocal imprinting, please include a brief discussion of studies of auditory perception using cross-fostered birds to examine innate preferences versus experience-dependent changes in auditory responses (e.g. Araki et al., 2017; Moore and Woolley, 2019).

3. In lines 187-189, please modify your description of hair cell regeneration as "a specialized avian feat". Sensory hair cell regeneration after antibiotic treatment has been shown in other vertebrates, including fish (in the inner ear and lateral line; reviewed by Monroe et al., 2015, Front Cell Neurosci), as well as invertebrates (e.g. frogs).

4. Please revise the section "Brains and genes for vocal learning" to either omit mention of selective auditory responses, or expand this section for clarity (lines 319-321). While many studies have shown that neurons in both the motor pathway and the anterior forebrain pathway respond selectively to playback of bird's own song in anesthetized birds, the role of such auditory responses is not entirely clear. Auditory responses in HVC and in LMAN and Area X are gated by behavioral state and are much less pronounced in awake birds (Schmidt and Konishi , 1998; Hessler and Doupe, 1999).

5. Please expand the discussion of the role of HVC in coding the timing of song (lines 323-327). For example, HVC neurons that project to RA fire once per motif while RA neurons may fire multiple bursts at specific times in song (Yu and Margoliash, 1996). Similarly, Long and Fee, 2008 is referenced, but it would help the reader to explain how the cooling experiments help to establish that HVC codes for time in song.

---

## [Author Response]

Summary:This article reviews the history of the zebra finch as a species of choice to study different aspects of developmental neurobiology and vocal communication. However, the article is lacking references to important works, and should also discuss the roles of the descending motor pathway in relation to the anterior forebrain pathway for song learning and adult song production.

We have now included a fully-cited section on the role of the motor pathway in mediating juvenile song learning and adult song production.

Essential revisions:1. Please include references to the work from labs that have substantially contributed to the fields of neuroscience and behaviour using zebra finches, including the Fee lab (MIT), the Goldberg lab (Cornell), the Sakata lab (McGill).

Dozens of previously unmentioned labs’ references and discussion of these works are now added.

Also please ensure that the article includes citations to the key works of other labs that are already cited, including the Jarvis lab (Rockefeller), the Long lab (NYU), the Brainard lab (UCSF), the Tchernikovski lab (CUNY) and the Theunissen lab (Berkeley). Please add up to 20 references to address this point.

Additional references (65 new ones) and discussion of these works are now also added to the manuscript.

Additionally, please consider the following:a) Please cite Desmond Morris (1954 and 1957) along with Immelmann when talking about the reproductive behaviour of zebra finches (line 35), and try to find a reference for Immelmann that is not an abstract for a conference.

Thank you. Done (Morris 1954) but Immelmann 1971 is a full article in the Proceedings of the 15^th^ IOC, it’s not just a conference abstract.

b) Please cite Katharina Riebel (2000) "Early exposure leads to repeatable preferences for male song in female zebra finches", or any of her later reviews, in addition to Braaten and Reynolds (1999) (line 40).

The Riebel 2000 paper and a Riebel-lab 2003 paper are now included.

c) Please check whether Day, 2019 is the correct citation in line 48; and whether work by the White, Mooney, Roberts or Scharff labs would be more appropriate when talking about genome editing.

Thank you. We have removed the incorrect Day et al. reference here and added additional avian genome editing review instead.

d) Please check whether there are references missing in line 51 after "illness".

Reference (Han and Park 2018, London 2020) is added.

e) Please add references for genetic diversity in line 73 (e.g. "Genetic variation and differentiation in captive and wild zebra finches (Taeniopygia guttata)", from the Forstmeier lab).

New Forstmeier lab references are added.

f) Please cite reviews on line 140, since the literature is extensive. "Sexual imprinting and evolutionary processes in birds: a reassessment", Ten Cate and Vos (1999), and "The impact of learning on sexual selection and speciation", Verzijden et al. (2012).

Two review papers are now also added as citations.

g) In line 145, Louder et al. covers auditory but not song production, please find an appropriate reference to cover both.

We are now citing an additional reference (Mooney 2009) to cover song production.

h) In line 320, please add a reference to Scharff's work on FoxP2

The Scharff TiG review/reference to FoxP2 is added as well as other Scharff lab references.

i) In line 334/35, please provide a reference for "the unique immune function of oscine birds inhibiting full viral delivery of gene constructs".

Reference of London 2020 is added.

2. Please add between 500 and 1000 words describing the roles of the descending motor pathway vis-à-vis the anterior forebrain pathway for song learning and adult song production; and the use of the zebra finch to study auditory perception of both biologically relevant stimuli (i.e. conspecific vocalizations) and others (e.g. human speech patterns).

We have now added discussion of both of these themes to the manuscript.

3. Figure 2B should include spectrograms of at least a female call (see line 272), perhaps both a male and female call, as well as the male song.

Call spectrograms are also added and turned into a new figure.

4. The authors' work currently makes up about a third of the references, please check whether these references are all necessary and whether it might be possible to replace some of them with references to other important works in the field.

We have added 65 new citations to the manuscript, only 59 of which were non-self citations, especially with respect to the neurobiological, developmental, and behavioral contents of the paper and so the proportion of self-citations is now much diminished (25% instead of the earlier 35%). However, we still note that almost exclusively the only field work conducted on wild zebra finches over the past 15 years has been through the lab of Prof. Simon Griffith, our senior author. The main focus of this body of work (of which we have only cited a small fraction) has been to provide a relevant behavioral and ecological context to inform the vast amount of captive work on this species.

[Editors' note: further revisions were suggested prior to acceptance, as described below.]

Summary:This article reviews the history of the zebra finch as a species of choice to study different aspects of developmental neurobiology and vocal communication, and particularly as a system for investigating sex differences in vocal communication. The revised manuscript is much improved, but it would still benefit of additional revisions to references, and including a discussion of studies of auditory perception using cross-fostered birds to examine innate preferences versus experience-dependent changes in auditory responses.

We now include more discussion of cross-fostering as a method to address innate vs. experience-dependent changes in auditory responses (something that I also did my postdoc research on!)

Essential revisions:1. Please make the following modifications to references:a) In line 37, add references to the work of the Fee and Yazaki-Sugiyama labs. These would be useful for neural mechanisms underlying sensorimotor learning (e.g. Okubo et al., 2015; Mackevicius, Happ, and Fee, 2020; Yanagihara and Yazaki-Sugiyama, 2019).

We have now added the content and the three references to the text.

b) In lines 38-39, the reference to Chen et al., 2017 does not seem to be appropriate, since that study focuses on vocal learning, not female preference. Please remove this reference and replace it with another that examines influences on female song preferences (e.g. Riebel, 2000).

This is not correct as Chen et al. 2017 does focus on experience-dependence in females’ song preferences for male song variants. We, therefore, kept this reference here: Chen Y, Clark O, Woolley SC (2017) Courtship song preferences in female zebra finches are shaped by developmental auditory experience. Proceedings of the Royal Society of London B 284: 20170054.

c) In lines 63 and/or 73, please add a reference to Cade et al., 1965, Water economy and metabolism of two estrildine finches, Physiological Zoology.

We have now added this reference to the first instance.

d) In lines 88-89, please add references to papers by Mets and Brainard (2018, 2019) that investigate the interaction of genes and social and/or auditory experience on vocallearning and performance. Please add a short discussion about gene-environment interactions that shape song learning based on the findings of these papers in the section "Brains and genes".

Thank you, we agree that G X E interactions are essential in this section and we added the more recent citation, of these two on Bengalese finches, from *eLife* to support our updated content for the requested new sentences regarding Gene-by-Environment interactions in song learning in zebra finches, too.

e) In line 159 please add a reference to Price, PH (1979) Developmental determinants of structure in zebra finch song. Journal of Comparative and Physiological Psychology.

We have now incorporated this reference twice at around this section.

f) Please check the references in lines 258-260. While several papers have shown that both calls and songs have strong individual signatures that can be used for individual recognition, the references cited did not specifically examine "high levels of coordinated duetting between the male and female".

Thank you. We have now rephrased this section and included our duetting reference here, too (Elie et al. 2010).

g) In lines 323-327, please add a reference to Aronov et al. (already cited in the article) in regard to the critical role of HVC for generating stereotyped, learned vocalization in lines 323-327.

We have now added this reference here, too.

h) In lines 330-331, please add references for the idea that the AFP generates song variability in juvenile birds (Olveczky et al., 2005, 2011; or a review article, e.g. Mooney, 2009; Woolley and Kao, 2015; or Dhawale, Smith and Olveczky 2017), and discuss these findings.

Thank you, we really needed references here. We have now cited two of these reviews and discuss briefly trial-and-error learning specifically in this section.

i) Please add references to studies showing that LMAN also sends an instructive signal that can bias song and drive systematic changes in mean pitch, mean duration and syllable transition probabilities (Turner and Brainard, 2007; Andalman and Fee, 2009; Ali et al., 2013, Charlesworth et al., 2012), and discuss these findings.

We have now added to our description and discussion of the role of LMAN and AFP in general, using the Andalman and Fee citation.

j) In lines 341-344, please add a reference for deafening experiments in juvenile birds, such as Price, 1979.

We have now added Price 1979 here, too.

k) In lines 349-351, it is unclear why the Ma et al., 2020 reference is included, since the paragraph discusses mechanisms by which evaluation of song performance using auditory feedback can drive changes in song. Ma et al., 2020 report activity prior to calls by conspecifics. While predictive activity may be important for learning or maintaining song, please make clear what the activity is predicting (timing of upcoming calls? Expected auditory feedback given the motor commands?) and how it relates to learning the tutor song model.

We have deleted the Ma et al. 2020 reference and the tangential text associated with it in this paragraph.

l) In line 401, please add Xiao and Roberts, Neuron, 2018 as a reference, in addition to Kearney et al., 2019.

We have now added this prior article here, too.

m) In lines 441-444, please add a reference to Shaughnessey et al., JCN, 2018 when mentioning the song system in females as that paper argues against the idea that females possess only vestiges of the song control network and suggests that the forebrain network may have a different function in females.

Thank you for this important reference, we now rephrased this sentence to be able to cite this article.

2. Given the manuscript's emphasis on vocal imprinting, please include a brief discussion of studies of auditory perception using cross-fostered birds to examine innate preferences versus experience-dependent changes in auditory responses (e.g. Araki et al., 2017; Moore and Woolley, 2019).

We now contrast female vs. male specific findings using these references in the paragraph just prior to the Conclusions.

3. In lines 187-189, please modify your description of hair cell regeneration as "a specialized avian feat". Sensory hair cell regeneration after antibiotic treatment has been shown in other vertebrates, including fish (in the inner ear and lateral line; reviewed by Monroe et al., 2015, Front Cell Neurosci), as well as invertebrates (e.g. frogs).

Thank you; we have corrected this flaw and have now referred to two additional citations reviewing (including the one suggested) this topic in non-birds, too.

4. Please revise the section "Brains and genes for vocal learning" to either omit mention of selective auditory responses, or expand this section for clarity (lines 319-321). While many studies have shown that neurons in both the motor pathway and the anterior forebrain pathway respond selectively to playback of bird's own song in anesthetized birds, the role of such auditory responses is not entirely clear. Auditory responses in HVC and in LMAN and Area X are gated by behavioral state and are much less pronounced in awake birds (Schmidt and Konishi , 1998; Hessler and Doupe, 1999).

We have now rephrased this section, and adopted the referee’s suggestion for modifying the text and for the two additional citations included.

5. Please expand the discussion of the role of HVC in coding the timing of song (lines 323-327). For example, HVC neurons that project to RA fire once per motif while RA neurons may fire multiple bursts at specific times in song (Yu and Margoliash, 1996). Similarly, Long and Fee, 2008 is referenced, but it would help the reader to explain how the cooling experiments help to establish that HVC codes for time in song.

Thank you for these suggestions; we have now included new text and the new citation (plus one more) regarding HVC, RA, and temperature-dependent effects on song production in this section.